



**Soil priming effects and involved microbial community along salt gradients**
Haoli Zhang[ab#], Doudou Chang[a#], Zhifeng Zhu[c], Chunmei Meng[b], Kaiyong Wang[a*]
[a] Agricultural College, Shihezi University, Shihezi 832003, China
[b] College of Land Science and Technology, China Agriculture University, Yuanmingyuan West Road,
Beijing 100193, China
[c] China National Seed Group Co.Ltd, Yazhou Science and Technology City, Sanya 572000, China
*All correspondence to:* Kaiyong Wang
Email: wky20@163.com
[#] Both equally contribute to this work
5 Figures
3 Tables
4 Supplementary Figures
text pages





**Abstract**
Soil salinity mediates microorganisms and soil process, like soil organic carbon (SOC) cycling.
Yet, how soil salinity affects SOC mineralization via shaping bacterial communities diversity and
composition remains elusive. Therefore, soils were sampled along a salt gradient (salinity at 0.25%,
0.58%, 0.75%, 1.00% and 2.64%) and incubated for 90 days to investigate i) SOC mineralization (i.e.
soil priming effects induced by cottonseed meal, as substrate) and ii) responsible bacteria community,
by using high throughput sequencing and natural abundance $^{13}$C isotopes (to partition cottonseed meal
derived $CO_2$ and soil derived $CO_2$. We observed negative priming effect during first 28 days of
incubation but turned to positive priming effect after day 56. Negative priming at the early stage might
be due to the preferential utilization of cottonseed meal. The followed positive priming decreased with
the increase of salinity, which might be caused by the decreased alpha diversity of microbial
community in soil with high salinity. Specifically, soil pH and EC along salinity gradient were the
dominant variables modulating the structure of microbial community and consequently SOC priming
(estimated by distance-based multivariate analysis and path analysis). By adopting O2PLS, priming
effects were linked with specific microbial taxa, e.g., Proteobacteria (*Luteimonas, Hoeflea* and
*Stenotrophomonas*) were the core microbial genus that attributed to the substrate induced priming
effects. Here, we highlight that the increase of salinity reduced the diversity of microbial community
and shifted dominant microorganisms that determined SOC priming effects, which provides a
theoretical basis for understanding of SOC dynamics and microbial drivers under salinity gradient.

Keywords: *Salt gradient, priming effects, bacterial community, core microorganisms*





## 1. Introduction

Soil organic carbon (SOC) is the largest pool (1500 Pg C) in the terrestrial carbon (C) cycle, and contains twice as much C as the atmosphere (Filley and Boutton, 2006; Wiesmeier et al., 2019). The input of substrate C can influence the output (i.e., $CO_2$ release) through a phenomenon called priming effect, which was firstly discovered by LÖhnis (LÖhnis, 1926). Substrate additions accelerate or decrease soil organic C mineralization, referred to positive or negative priming effects (Kuzyakov et al. 2000). The intensity of the priming effect affects the turnover of SOC and thus storage pool (Sullivan and Hart, 2013). Soil priming effects are affected by many biotic and abiotic factors (Lavelle, 1997; Martin W, 2019), to investigate abiotic and biotic mechanisms underlying SOC priming enhance strong understanding of the SOC cycling.

Soil priming effects is affected by soil fauna animals (Scheu and Parkinson, 1994), activities, diversity and composition of microbial community (Di Lonardo et al., 2017; Fontaine et al., 2011). The microbial decomposers are the major player in the decomposition process of added C sources. The addition of substrate, such as composts (Xun et al., 2016), animal sludges (Hartmann et al., 2015), sewage sludges (Su et al., 2017; Wagner and Raquel, 2011) and plant residues (Dai et al., 2017), generally increases soil microbial biomass C and stimulates the microbial activities thus enhanced the loss of SOC (positive priming effects) (Fontaine et al., 2003; Bird et al., 2011; Li et al., 2018; Ali et al., 2019).

Concerning abiotic factors, the priming effect can be controlled by climate variables (Hagemann, 2008), and soil properties, like pH, EC, TN, etc (Blagodatskaya and Kuzyakov 2008; Luo et al., 2017). To understand how environmental and edaphic factors affect the processes of SOC mineralization, is important to estimate terrestrial C pool (Lehmann and Kleber, 2015). Although many studies have tested the effects of soil pH, SOC content, and other edaphic variables on soil priming effect, few study investigated soil priming effects in salinity soil (Asghar et al., 2012), especially linked with soil microbial community structure and their functions in C decomposition (Soina





et al., 2018).
Soil salinization is an increasing environmental problem caused by natural and
human activities in the arid and semi-arid area (Wichern et al., 2006). Salinization is
often a major threat to crop productivity in agricultural land. Soil microorganisms suffer
from osmotic stress. Soil salinity often cause microbial death or dormant. It was widely
reported that the increased salinity decrease microbial biomass, enzymatic activity, and
alpha diversity of microbial community (Laura, 1974; Pathak and Rao, 1998; Rietz and
Haynes, 2003). Soil salinity is reported to the major determinants of composition,
activity of microbial community (Kamble et al., 2014). Although salinity is reported to
be a vital factor in influencing microorganisms in the arid and semi-arid area, limited
studies investigated C processes (e.g. priming effect) driven by microbial community
in salinity soils (Sardinha et al., 2003).
Thus, we sampled the soils along natural salinity gradients (0.25%, 0.58%, 0.75%,
1.00%, 2.64% apart from total water-soluble salt). Based on these soils, we conducted
a 90 days of indoor incubation applying C3 substrate of cottonseed meal ($\delta^{13}C$=-
23.47‰) to C4 soils with salt gradient ($\delta^{13}C$ between -14.21‰ and -16.01‰), to
investigate: 1) mineralization rate of cottonseed meal and induced soil priming effects
along salt gradients; 2) diversity of microbial community in the soils with increased
salinity, and 3) identify the bacteria taxa associated Soil priming. We hypothesized that
i) soil microbial community diversity and composition will be different with the
different in soil variables particularly pH and EC along salinity gradients, and ii) Soil
C processes like priming effects will be regulated mainly by microbial community and
especially the core microbial species. To clarify the priming effects and involved
microbial groups would help us better understanding C sequestration potential and
underlying mechanisms in saline soils.

**2.    Materials and methods**
*2.1. Soil sampling and cottonseed meal production.*


The soil type was gray desert soil, which was collected from farmlands (82.90°
longitude, 44.96° latitude) in Bole City, Bortala, northern Xinjiang Uygur Autonomous
Region, northwest China. The farmlands soil is naturally formed original saline-salinity
soil and with a continuous 30 years planting of maize (C4 crop) and maize straw
returning to soil for 7-8 year. The soil samples were indoor air drying and hand-picked
to remove visible other debris, animal and plant residues and then sieved at field
moisture (<2mm) and subsequently adjusted to 40% of water holding capacity (WHC).
Texture was determined by the pipette method without carbonate in all soil samples.
They were then incubated at 25 ℃ for 7 days before starting the experiments, to allow
any early sampling and sieving effects to subside.
Cottonseed meal is a kind of reddish or yellow granular material obtained by
pressing, leaching and other cottonseed. The cottonseed meal was purchased from the
market and dried at 105 °C for 24 h indoor, then further pulverized by a ball mill and
passed through < 2 mm sieve.

*2.2. Soil and substrate analyses*
EC and pH of soil and cottonseed meal were measured at a soil: water ratio of 1:5
(weight/weight). Air-dry soil (5 g, <2 mm) and 25 ml of deionised water were shaken
together for 1 min and left to settle for 30 min, which was repeated once more before
pH was determined with a pH electrode. Soil water-soluble salt was analyzed by
weighted at a soil:water ratio of 1:5 (weight/weight). Air-dry soil (5 g, <1 mm) and 25
ml of deionised water were shaken together for 30 min, filtration to obtain clear filtrate,
using thermostat water bath to evaporate and weigh. Total soil C and N concentrations
(air-dried, milled <150 μm) were determined by dry combustion (LECO CNS 2000,
LECO Corporation, Michigan, USA). Soil microbial biomass C was determined by
fumigation extraction (Vance et al., 1987; Wu et al., 1990). The $K_2SO_4$ extractable
organic C was determined using an organic carbon autoanalyser (Shimadzu, Analytical



Sciences, Kyoto, Japan). Soil microbial biomass C (Bc) was calculated from: Bc = 2.22
Ec, where Ec = [(organic C extracted from fumigated soil) minus (organic C extracted
from non-fumigated soil)]. The natural $\delta^{13}C$ (‰) abundance of the soils (air-dried,
milled <200 µm) was determined using an elemental analyser-isotope ratio mass
spectrometer (Sercon Ltd, Crewe, UK). All measurements are given on an oven-dry
weight basis (o.d., 105 °C, 24 h).

The $\delta^{13}C$ (‰) abundance of the cottonseed meal (air-dried, milled <200 µm) was

determined using an elemental analyser-isotope ratio mass spectrometer (Sercon Ltd,
Crewe, UK). The main elemental composition of the substrate was determined using
elemental analysis (Vario EL Cube, Hanau, Germany), with the samples combusted at
1200 °C. Natural $\delta^{13}C$ (‰) abundance ,the total carbon, total nitrogen contents and C/N
of the cottonseed meal was presented in Table 1.

*2.3.   Experimental design*

After pre-incubation, five soils with salinity gradient were thoroughly mixed with

cottonseed meal at 20 mg C $g^{-1}$ soil (d.w. basis), and incubated over 90 days following
moisture adjustment to 40% of water-holding capacity (WHC) to investigate the
substrate mineralization and priming effects. Each soil sample (40 g d.w. basis) was
incubated in a 100 ml beaker inside a 1 L brown glass jar. Three jars with only water
and NaOH were set as blank. All the jars were sealed with a rubber bung and incubated
in a randomized block design at 25 °C for the 90 days of incubation. The NaOH vials
were changed after 1, 3, 5, 7, 14, 28, 56 and 90 days for determination of evolved $CO_2$
and $^{13}C$–$CO_2$ (‰). Meanwhile, soil biomass C, $NH_4^+$, $NO_3^-$, pH, EC, TC, TN and DNA
extraction were measured at day 28.

*2.4.   Soil $CO_2$-C and its isotopic composition*





Soil C evolved as $CO_2$-C in jars was measured by trapping $CO_2$ in 1 M NaOH

(20 ml) during soil incubation. After the NaOH (20 ml) trapping $CO_2$ at different
periods of soil incubation, 5 ml 1 M NaOH of each sample was mixed with 10 ml
deionised water and titrated with 0.05 M standardised HCl by the TIM840 autotitrator
(Radiometer Analytical, Villeurbanne Cedex, France). Meanwhile, the $\delta^{13}C$ (‰) of
trapped $CO_2$-C was precipitated, with 8 ml of the 1 M NaOH (20 ml) mixed with 8 ml
1.5 M $BaCl_2$ in vials (Aoyama et al., 2000). The $BaCO_3$ precipitate was trapped on the
glass fibre the filter, rinsed with deionised water several times, and dried overnight
(80 °C), weighed (0.100-0.200 mg) into tin capsules, and analyzed for $\delta^{13}C$ on an
elemental analyzer-isotope ratio mass spectrometer (Sercon Ltd, Crewe, UK).

*2.5. DNA exaction and sequencing*

The total soil DNA was extracted from 0.50 g of moist soil using a FastDNA Spin

Kit (MP Biomedicals, Santa Ana, CA, USA) according to the manufacturer's protocol.
The extracted DNA was dissolved in 50 µl of TE buffer, quantified using a
spectrophotometer and stored at −20 °C until sequencing.

V3-V4 hypervariable regions of the bacterial 16S rRNA gene were amplified with

primers   341F   (5'-CCTAYGGRBGCASCAG-3')   and   806R(5'-
GGACTACHVGGGTWTCTAAT-3'). The PCR reactions were conducted with a
thermocycler PCR system (GeneAmp 9700, ABI, USA) by using the following
programs: 3 min of denaturation at 95 °C; followed by 27 cycles of 30 s at 95 °C, 30 s
at 55 °C, and 45 s at 72 °C; and a final extension at 72 °C for 10 min with a thermocycler
PCR system (GeneAmp9700, ABI, USA). PCR amplicons pooled from the triplicate
reactions were purified using a QIAquick PCR purification kit (Qiagen, Shenzhen,
China), and quantified using a NanoDrop ND-1000 spectrophotometer (Thermo
Scientific, Waltham, MA, USA). The PCR products were purified, mixed, and sent to
Majorbio, Inc. (Shanghai, China) for sequencing based on the Illumina MiSeq platform.



*2.6.   Calculations*
2.6.1. *$CO_2$-$\delta^{13}C$ emission*
The mineralisation of cottonseed meal was separated from SOC mineralisation
according to the change of stable isotopic composition ($\delta^{13}C$) with time. The standard
equation for determining $\delta^{13}C$ (‰) is derived from:
$\delta^{13}C$ (‰) = [($R_{sample}$/$R_{VPDB}$) − 1] × 1000,               Eqn. 1
where $R_{sample}$ is the mass ratio of $^{13}C$ to $^{12}C$ of each sample and $R_{VPDB}$ is the
international PDB limestone standard. The labeled $^{13}C$ (%) of cottonseed meal was then
estimated from:
$CO_2$-$^{13}C$ (%) = ($\delta_{treatment}$- $\delta C4$) / ($\delta C3$ - $\delta C4$),             Eqn. 2
where $CO_2$-$^{13}C$ (%) is the proportion of evolved $CO_2$ from C3 (cottonseed meal)
matter, $\delta_{treatment}$ is the $\delta^{13}C$ (‰) in treatments of soil with cottonseed meal, $\delta C4$ is the
$\delta^{13}C$ (‰) in control soil and $\delta C3$ is the $\delta^{13}C$ (‰) from cottonseed meal. Thus, the $CO_2$-
C produced from cottonseed meal during the incubation was calculated from:
$CO_2$-$^{13}C$ ($\mu g\ g^{-1}$ soil) = $CO_2$-$^{13}C$ (%) × total $CO_2$-C ($\mu g\ g^{-1}$ soil)/100,      Eqn. 3

$CO_2$ from SOC was $CO_2$-$^{13}C$ subtracted from total evolved $CO_2$-C. The absolute
soil priming effect (or primed soil $CO_2$-C) with the addition of cottonseed meal was
calculated from:
Primed soil $CO_2$-C ($\mu g\ C\ g^{-1}$ soil) = $CO_2$-$C_{treatment}$ - $CO_2$-$C_{control}$         Eqn. 4
where $CO_2$-$C_{treatment}$ is the non-isotopically labeled $CO_2$-C evolved from
cottonseed meal amended soil, $CO_2$-$C_{control}$ is non-isotopically labeled $CO_2$-C evolved
from soil without cottonseed meal.

*2.7. Statistics*
The data of 16S gene sequencing were processed using the Quantitative Insights
Into Microbial Ecology (QIIME) 1.9.0-dev pipeline (Caporaso et al., 2010). In brief,
Reads with less than length 200 bp and ambiguous bases were discarded. The sequences





were then binned into operational taxonomic units (OTUs) by UCLUST (Edgar, 2010)
based on 97% pairwise identity. Chimeric OTUs identified by USEARCH (Edgar et al.,
2011) in QIIME were removed. The most abundant sequence from each OTU was
selected to represent that OTU. Taxonomy was assigned to 16S OTUs against a subset
of the Silva 104 database. The representative OTU sequences were aligned using
PyNAST (Caporaso et al., 2010). We obtained between 64,425 and 89,989 clean_reads
per sample for all experimental samples.
To avoid potential bias caused by sequencing depth, all sample datasets were
rarefied for the bacteria α-diversity and β-diversity analyses. Faith's phylogenetic
diversity was calculated to provide an integrated index of the phylogenetic breadth
across taxonomic levels (Faith, 1992). To compare β-diversity between samples,
principal coordinate analyses based on the unweighted and weighted UniFrac
(Lozupone et al., 2007a) distances were calculated using the function 'pcoa' in the R
package 'Ape'. Additionally, permutational multivariate analysis of variance
(PERMANOVA) was carried out using the function 'adonis' in the R 'vegan' to
measure effect size and significance on β-diversity. The variable influence projection
(VIP) value was processed using the way of O2PLS analysis by the SIMCAP 14
(Version 14.1.0.2047) (Wang et al., 2016). The y-matrix was defined as the
environmental factors datasets and the x-matrix was defined as the microbial
community on genus level dataset.
Data were logarithmically transformed and analyzed by ANOVA. All analyses
were performed using SPSS software (13[th] edition). Pearson's correlation analyses were
performed to assess the linear correlation among soil physio-chemical properties and
microbial community. MULTIVARIATE analysis were operated to investigate
interaction of salinity treatments on bacteria community parameters.

**3.    Results**
3.1. Soil physicochemical properties along salt gradients





The major soil physicochemical properties along salt gradients were presented
(Table 1) and all of soil physicochemical properties has significant difference (P < 0.05).
The total soluble salinity content in the soils ranged from 0.25% to 2.64% of salinity
soils, soil salt gradients increasing gradually from salinity 1 samples to salinity 5
samples. The pH and EC in soils ranged from 8.45 to 8.85 and from 1.06 ms cm$^{-1}$ to
7.75 ms cm$^{-1}$. Soil total C and N were increased with salinity, ranging from 3.16% to
3.57%, and from 0.18% to 0.26%. The $\delta^{13}$C value for soils are between -14.21‰ and -
16.01‰, which were relatively enriched compared to cottonseed meal (-23.47‰). This
allowed separation of soil derived $CO_2$ from total evolved $CO_2$, according to the classic
mixed modeling.

3.2. Total $CO_2$ evolution
During the whole 90 days of incubation, the cumulative $CO_2$ evolved had similar
trends, which the amount of $CO_2$ increased with the incubation times (Fig. S1). The
cumulative $CO_2$ evolved increased more rapidly with the addition of cottonseed meal
before 14 days, compared to non-amended soils. At 90 days of incubation. The
cumulative $CO_2$ evolved in the soil with the lowest salinity (Salinity 1) gave the lowest
CO2 emission (597 μg C g$^{-1}$) in the non-amended soils (Fig. S1, P < 0.001).

3.3.    Cottonseed derived $^{13}CO_2$ and soil priming effects
The total cumulative $CO_2$-C was divided three parts based the $\delta^{13}$C value,
including basal soil-derived $CO_2$, cottonseed meal-derived $CO_2$ and primed soil $CO_2$
(Fig.1). The cottonseed meal-derived $CO_2$ had a significant contribution to the total $CO_2$
evolved during the early incubation period. The cottonseed meal-derived $CO_2$ was
significantly higher in Salinity 1, Salinity 2 and Salinity 3 than in Salinity 4 and Salinity
5 before 28 days incubation. Meanwhile, the soil priming effects was negative in all
amended soil treatments before 28 days incubation and the direction of priming effect
in most of soil samples turned into positive after 28 days. During the whole 90 days





incubation, there was a negative correlation between cottonseed meal-derived $CO_2$ and
primed soil $CO_2$   (Fig. 2).

3.4. Bacterial diversity and community structure

The number of sequences ranged from 64,425 to 91,261 for per sample (average

valve of 80,602). About 27,990 OTUs in total were obtained under different five
treatments. Bacterial community diversity was measured by a series of OTU-based
analyses of alpha diversity including chao1 estimator, and observed_species   in the
QIIME pipeline (Fig. 3). Chao1 diversity estimator and observed_species was
significantly different in treatments, being the highest in Salinity 1, followed by Salinity
3, Salinity 2, Salinity 4 and Salinity 5 ($P < 0.01$). In general, bacterial community
diversity decreased with increasing salinity (Fig. 3).

The most abundant phylum in the soils and their correlation with salinity were

shown in Fig. 4. Among them, Actinobacteria was the dominant taxa in all soils, with
the abundance ranging from 50.07 % (Salinity 3) to 68.99 % (Salinity 4). the relative
abundance of Bacteroidetes, Firmicutes, and Deinococcus-Thermus   increased with
the salinity, while Acidobacteria decreased with salinity degree.

Based on OTUs of five gradient salt treatments, the PCA analysis showed that

treatments from Salinity 2 and Salinity 4 clustered together. Meanwhile, soil samples
of Salinity 1, Salinity 3 and Salinity 5 distributed in the first, fourth and three quadrant,
which indicated that these treatments had large environmental heterogeneity (Fig. S4).

In order to visualize the relationship between environmental factors and microbial

community, *Canonical Correspondence Analysis* (CCA) was conducted, showing that
$NO_3^-$-N, EC and TC had a more obvious impact than other factors for microbial
community (Fig. 3). Soil EC were positively correlated with pH, $NH_4^+$-N, and
negatively correlated with TN, TC and MBC. Mantel test and Distance-based
multivariate analysis showed the contribution rate of different environmental factors





account for 78% of the variability of microbial communities (Table 2). The value of pH
(31%) and EC (12%) had a strong influence on microbial community.

3.5. Relation between soil microbial community and C dynamics

Based on the O2PLS analysis, the variable influence projection (VIP) values of

bacterial genus more than 1.00% were showed their contributions to C decomposition
of cottonseed meal-derived C, basal soil-derived C, and primed soil C (Table 3). There
were many microbial taxa positively correlating to soil primed $CO_2$, for insatnce, genera
of Actinomarinales, Luteimonas, Nocardioides, Hoeflea, Intrasporangium, Nitrolancea,
Pseudarthrobacter and Stenotrophomonas had a positive correlation with primed $CO_2$.
In order to further to evaluate the relationship between soil properties, soil bacterial
communities and C decomposition, we used the structural equation modeling (SEM) to
suggest the direct and indirect impacts of salinity and microbial community on soil C
decomposition (Fig. 7). The result showed that soil pH and EC had negative
contribution to bacterial diversity, while bacterial diversity had a strong positive
influence on the primed soil C (Fig. 5). For instance, salinity properties of EC had a
directly negative influence on the bacterial diversity but positive influence on the
primed soil C. Meanwhile, pH were negatively correlated with bacterial diversity and
positively correlated with substrate derived C.

**4. Discussion**
4.1. Soil priming effects along salty gradients

Understanding soil C dynamics along salinity gradients is crucial to predict C

sequestration in salty soils. In the early stage of the incubation, we observed that the
cumulative substrate derived $CO_2$ in the soils with lower salinity was significantly
higher than soils with higher salinity (Fig. 1), which can be possibly explained by that
high salinity inhibited microbial activity. Many studies have reported the influence of
soil salinity on organic matter decomposition, mostly, the decomposition of organic



matter are decreased by salinity (Wichern et al., 2006; Ghollarata and Raiesi, 2007;
Tripathi et al., 2007; Setia et al., 2012). Yet, the response of microbial community to
the increasing levels of salinity and consequent effects on soil priming effects remains
largely unknown.
Here, we found soil priming effects was gradually changed from negative to
positive priming effect (Fig. 1). The early pattern of the dynamics of the priming effect
in this study was similar to other studies showing preferential utilization of labile C
substance. The first phase of negative priming effects was likely to be caused by
microbial assimilation of substrate. The soil microbes turned to use the new added
substrate and thus used less of the original SOC. This was attributed to "preferential
substrate utilization" (Perelo et al., 2005).
Soil microbial biomass-related growth predominating in the first phase were most
likely to utilize SOC, leading to a positive priming effects after substrate was largely
vanished. The magnitude of priming effects depends on soil microbial biomass size
(Schneckenberger et al., 2008). It was found that the amount of added easily available
organic C is beyond 50% of microbial biomass C (Blagodatskaya and Kuzyakov, 2008).
Namely, the second phase of positive PEs probably was due to increased biomass size
and enhanced demand on SOC. Secondly, C that was assimilated into microbial
biomass in the first stage may also be mineralized in the second stage due to the
turnover of microbial biomass (Shahbaz et al., 2017; Perelo et al., 2005).

4.2. Microbial community along salt gradients
Previous studies concerning the impact of salinity on soil microbial community
used different soils with a range of salt levels. In the present study we investigated the
influence of soil salinity on microbial communities in soils from the closed area
covering a range of salt content. Similarly, Rousk et al. (2011) also used agricultural
soils from the same area representing a range of soil salinity. Here, we found microbial
diversity (alpha diversity) decreased with increasing salinity (Fig. 3). The negative



impact on microbial diversity can be explained by that the accumulation of large
amounts of salt in the soil raised the extracellular osmotic concentration (Rath and
Rousk, 2015; Oren, 2011). The high osmotic pressures made it difficult for many
microorganisms to adapt to and thus reduce their biological activity. The changes of
soil microbial community structure were also explained by salinity (Herlemann et al.,
2011; Campbell and Kirchman, 2013). We found that Bacteroidetes, Firmicutes,
Acidobacteria and Deinococcus-Thermus were dominant in these soils (Fig. 4). These
results are supported by previous findings that Firmicutes possess the high salinity
resistance. Other studies also found that Bacteroidetes is dominant taxa in alkaline
saline soil because of its resistant to salt (Valenzuela-Encinas et al., 2009; Keshri et al.,
2013). Other study shows that the dominant phyla are Bacteroidetes and followed by
Proteobacteria in the haloalkaline soil (Keshri et al., 2013). These results are consistent
with the esuarine or marine environments, despite some studies suggest that soil salinity
is not found to be a decisive factor for bacterial community and their growth (Rousk et
al., 2011).
The difference of microbial community structure is affected by many soil variables,
and pH and EC were the most important ones (Fig. 3; Table 2). Our results showed that
the value of soil pH and EC would significantly affect the microbial community
structure and the combined contribution rate of these two variables to microbial
community was 43% (Table 2). At high levels of salt and alkaline arid condition, soil
pH has been also shown to have a very powerful influence on the soil bacterial
community structures (Bååth and Anderson, 2003; Fierer and Jackson, 2006; Rousk et
al., 2010). Meanwhile, it is consequently unlikely that soil pH differences between the
studied soils obscured the influence of salt (Rousk et al., 2011). Salinity has been
identified as one of the most potent environmental factors that determine assembly of
microbiome. Salinity has been regarded to play the vital role in shapiong microbial
community in different ecosystem. This, despite the clear evidence from aquatic





microbial ecology (Lozupone and Knight, 2007b), show a potential for salt to affect
soil microbial communities apart from that of pH (Rath and Rousk, 2015).

4.3. The core microbial taxa regulating C decomposition along salinity gradient
The correlation of microbial taxa and SOC decomposition (priming) were found
according to the results of O2PLS and SEM (Table 3; Fig. 5). Here we showed that
*Streptomyces* (Actinobacteria), Glycomyces (branch of Actinobacteria), Agromyces
(branch of Actinobacteria), and Sphingomonas (branch of Proteobacteria) at the genus
level were significantly correlated with the C process particularly primed soil-drived C.
Most of these functional taxa belonged to Actinobacteria and Proteobacteria. In a recent
study, Ren et al. (2018) found that Actinobacteria had negative impact on SOC
mineralization across land-use change (Fierer et al., 2007; Goldfarb et al., 2011) and
Proteobacteria drove the positive soil respiration (He et al., 2012; Stevenson et al.,
2004), indicating the balance of soil C dynamics were largely regulated by these two
phyla. We found similar result that Streptomyces (branch of Actinobacteria) had a
negative correlation with primed soil $CO_2$. Actinobacteria are able to grow
preferentially on the C-rich refractory materials and relatively easily decompose the
cellulose, lignocellulose (Khodadad et al., 2011), indicating these microorganisms
preferentially use the C source that is used partially by others.
Although some studies suggest soil salinity may not be a vital factor for C
decomposers (Rousk et al., 2011), the composition of microbial community are
considered to play a decisive role in determining C dynamic processes in response to
salt stress (Ramsey et al., 2005; Schimel et al., 2007; Nottingham et al., 2009). Here,
SEM analysis showed that soil pH and EC in salted soils reduced microbial diversity
and thus limited the utilization of SOC by microbial community, It was reported that
high pH and salinity are the major determinants of soil microbial activity and
community structure (Kamble et al., 2014).





**5. Conclusion**
Soil priming effect turned from negative to positive at the later stage of incubation
(day 28), because microorganisms turned to decompose SOC from the labile substrate.
With the increase of salinity, the diversity of microbial community decreased. Soil
microbial community was mainly controled by soil pH and EC. By O2PLS, we found
Actinobacteria and Proteobacteria (*Luteimonas, Hoeflea* and *Stenotrophomonas*)
dominant in these soils were the core microbial taxa that affecting the process of organic
C mineralization, particularly soil primed $CO_2$.

**Acknowledgements**
This study was supported by the Special Fund for Key Science & Technology
Program in Xinjiang Province of China (No. 2022B02021-3-1) .

**Data availability**
The datasets used and analysed during the current study available from the
corresponding author on reasonable request.

**Author contributions**
K.W. conceptualized and conducted the experiment. H.Z. and D.C. conducted the
data analysis and wrote the manuscript, conducted the indoor experiment. C.M. and
Z.Z. assisted in conducting the experiment. All authors reviewed the manuscript.All
authors contributed to the manuscript and approved the submitted version.

**Competing interests**
The authors declare no competing interests.

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





**Table 1.** Soil samples and Cottonseed meal properties

|  | **Salinity 1** | **Salinity 2** | **Salinity 3** | **Salinity 4** | **Salinity 5** | **Cottonseed meal** |
|---|---|---|---|---|---|---|
| **Total C (%)** | 3.38b | 3.18c | 3.16c | 3.57a | 3.35b | 42.98 |
| **Total N (%)** | 0.18d | 0.19d | 0.20c | 0.22b | 0.26a | 5.84 |
| **C/N ratio** | 18.32a | 16.56b | 15.71c | 16.54b | 12.94d | 7.38 |
| **$\delta^{13}C$ value (‰)** | -14.21a | -14.79c | -14.60b | -14.55b | -16.01d | -23.47 |
| **pH ($H_2O$)** | 8.85a | 8.45c | 8.58b | 8.59b | 8.55b | 7.63 |
| **EC (dS m$^{-1}$)** | 1.06e | 1.96c | 1.28d | 2.64b | 7.75a | 2.56 |
| **Salinity (%)** | 0.25e | 0.58d | 0.75c | 1.00b | 2.64a | ND |







**Table 2.** Mantel test and Distance-based multivariate analysis relevance and
contribution rate between soil properties and bacterial community compositions.

|  | pH | EC | $NO_3^--N$ | $NH_4^+-N$ | MBC | TN | TC |
|---|---|---|---|---|---|---|---|
| Correlation | 0.74** | 0.56** | 0.36** | 0.68** | 0.31** | 0.11 | 0.27 |
| Contribution | 0.31** | 0.12** | 0.05 | 0.04 | 0.16 | 0.03 | 0.07** |

Note:* $p < 0.05$, ** $p < 0.01$






**Table 3.** The variable influence projection (VIP) value and Spearman's correlation between the relative abundances of genera and C dynamic.

| Phylum-Genus | VIP | Cottonseed meal $CO_2$-C($\mu$g g$^{-1}$) | Primed soil $CO_2$-C($\mu$g g$^{-1}$) | Basal soil $CO_2$-C($\mu$g g$^{-1}$) |
|---|---|---|---|---|
| Actinobacteria-Actinomarinales | 1.36 | | 0.63** | |
| Proteobacteria-Luteimonas | 1.31 | | 0.80** | |
| Actinobacteria-Nocardioides | 1.30 | | 0.54* | |
| Proteobacteria-Hoeflea | 1.29 | | 0.73** | |
| Actinobacteria-Streptomyces | 1.27 | | -0.84** | |
| Actinobacteria-Glycomyces | 1.26 | 0.63** | | |
| Actinobacteria-Marmoricola | 1.26 | -0.52 | | |
| Proteobacteria-Nitrosospira | 1.23 | | 0.59 | |
| Actinobacteria-Intrasporangium | 1.22 | | 0.60* | |
| Actinobacteria-Agromyces | 1.19 | | | 0.58* |
| Proteobacteria-Sphingomonas | 1.18 | | | 0.65** |
| Actinobacteria-Myceligenerans | 1.16 | | | |
| Chloroflexi-Nitrolancea | 1.15 | | 0.65** | |
| Actinobacteria-Pseudarthrobacter | 1.06 | | 0.62** | |
| Proteobacteria-Stenotrophomonas | 1.00 | -0.50 | 0.72** | |

Note:* $p < 0.05$, ** $p < 0.01$





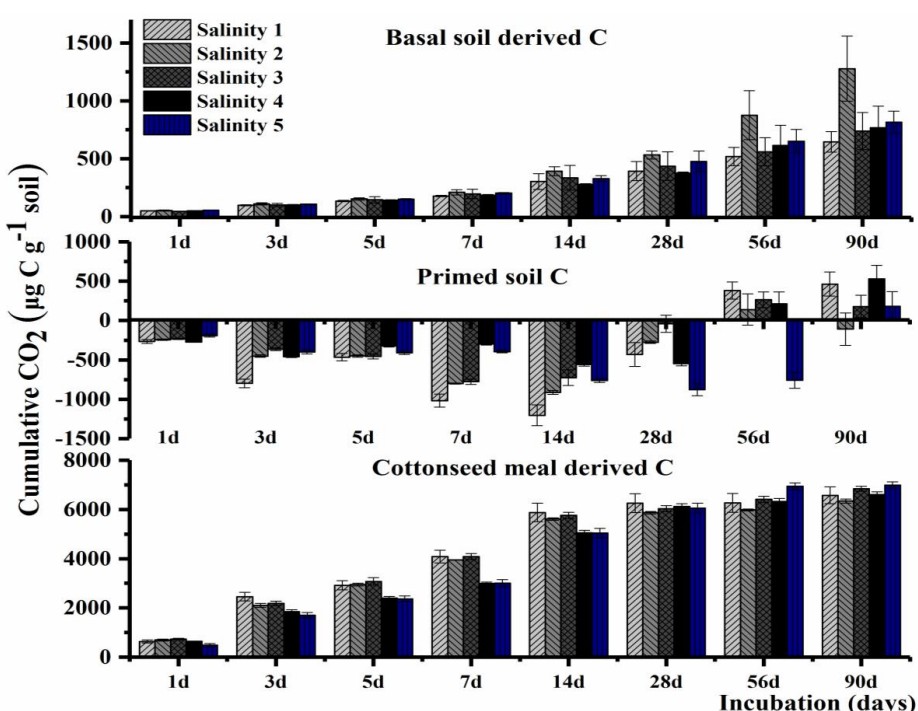

652

**Fig. 1.** Partitioning of $CO_2$ evolution after addition of cottonseed meal in different five salinity soils. Cumulative $CO_2$ evolved from salinity soil of 0.25 % (a) , 0.58 % (b) , 0.75 % (c) ,1.00% (d) and 2.64%(e) . Error bars represent standard errors of the means (n = 3).







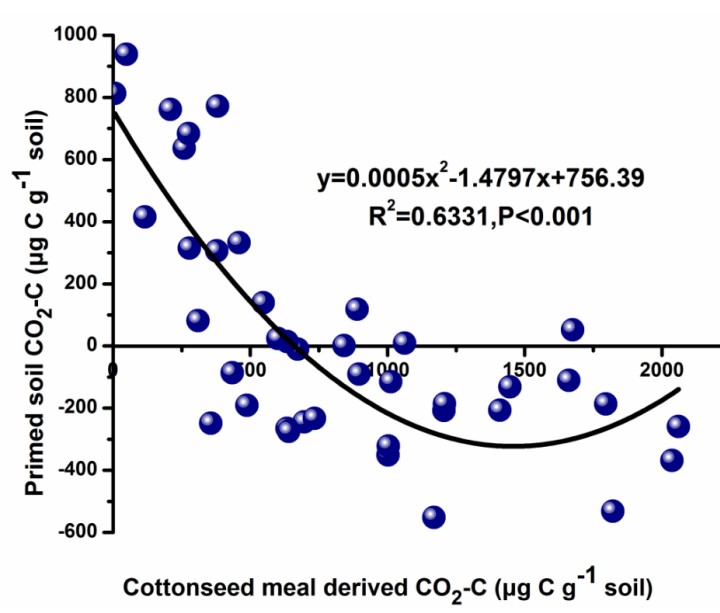


**Fig. 2.** Correlation between primed soil mineralisation and cottonseed meal
mineralisation following different five salinity soils during 90 days incubation






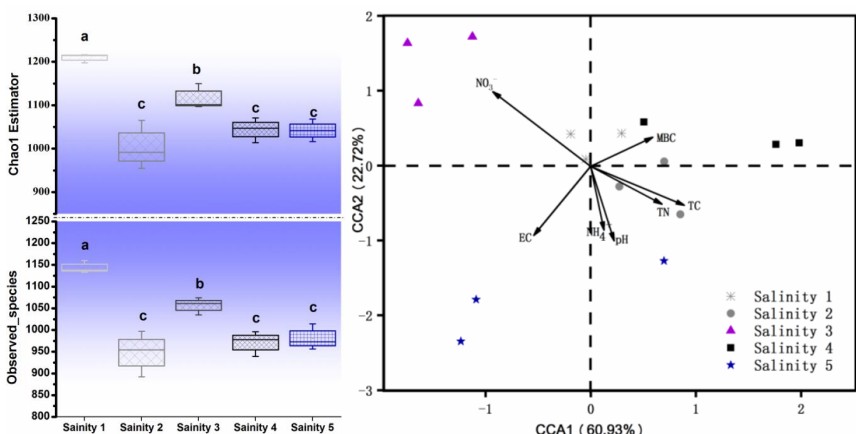

**Fig. 3.** Microbial community alpha diversity (Chao1) observed_species and beta diversity. Within each panel, boxplot data refer to maximum date (top line), 99%(the second line),mean (the third line), 1% (the fourth line) and minimum date (bottom line) of the different treatments, with statistical significance ( $P < 0.05$).




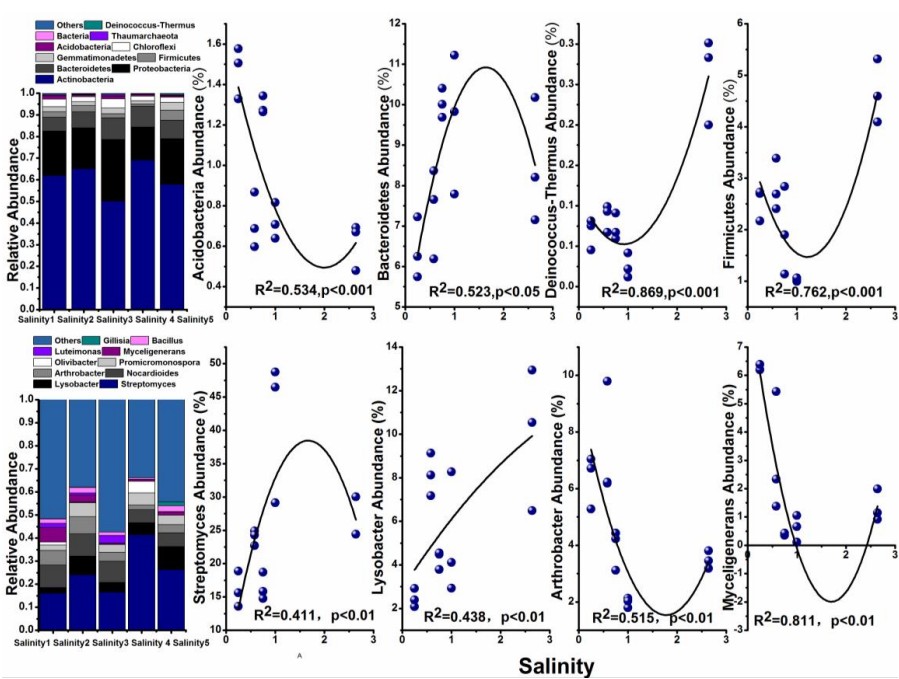

**Fig. 4.** The top 10 of phylum in bacterial community in soils with a gradient of salinity





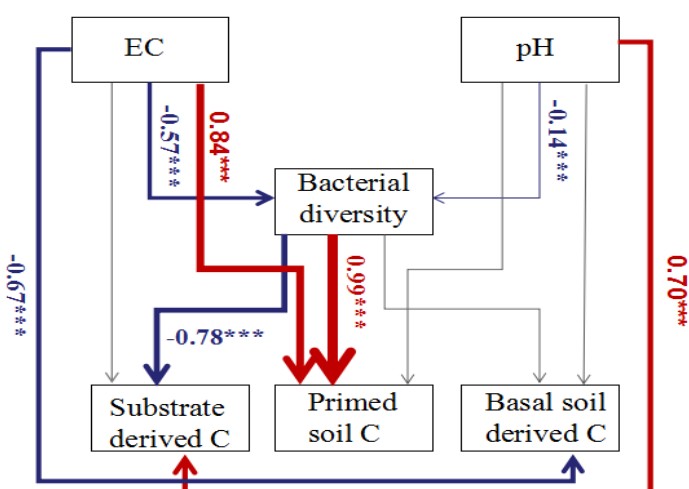

$$\chi^2 = 0.85, \; P = 0.65, \; GFI = 0.98, \; RMSEA < 0.001$$

**Fig. 5.** Path analysis detecting the underlying causal relationships between soil salinity physicochemical factors and microbial community composition of carbon dynamics in the soilt system. Red lines indicate positive relationships, while blue lines indicate negative relationships. The width of arrows indicates the strength of significant standardized path coefficients ($P < 0.05$). Paths with non-significant coefficients are presented as gray lines. ***$P < 0.001$; **$P < 0.01$; *$P < 0.05$