# Peer review of "Soil priming effects and involved microbial community along salt gradients 1 2 Haoli Zhangab#, Doudou Changa#, Zhifeng Zhuc, Chunmei Mengb, Kaiyong Wanga\* 3 4 a Agricultural College, Shihezi University, Shihezi 832003, China 5 b College of Land Science and Technology, China Agriculture University, Yuanmingyuan West Road, 6 7 Beijing 100193, China cChina National Seed Group Co.Ltd, Yazhou Science "

_Biogeosciences, 2023_

## Referee Comment (RC2)

Thank you for your advice. I will answer and revise the following questions.

1.  Materials and methods section 2.1 Soil sampling and cottonseed meal production does not provide a detailed description of the sampling site and does not mention the characteristics of the local soil type, etc. Please add.

    **Response:** Many thanks for your review and suggestions. Please see lines 96-100 "The soil type was gray desert soil, which was collected from farmlands (82.90° longitude, 44.96° latitude) in Xiao Yinpan town,Bole City, Bortala, northern Xinjiang Uygur Autonomous Region, northwest China. The farmlands soil is naturally formed original saline-salinity soil and with a continuous 30 years planting of maize (C4 crop) and maize straw returning to soil for 7-8 year. " And the cottonseed meal productions see the lines 106-109 and table 1.

2.  What is the season for the collection of agricultural soil samples? Is it an inter-root or non-root soil? What is the soil's mode of collection? Please add.

    **Response:** We have revised this sentence following your suggestion. Please see lines 101-107 "In September 2021, we determining the sampling area, and use the five-point sampling method to collecting non-rhizosphere soil.  The soil samples were indoor air drying and hand-picked to remove visible other debris, animal and plant residues and then sieved at field moisture (<2mm) and subsequently adjusted to 40% of water holding  capacity (WHC). Texture was determined by the pipette method without carbonate in all soil samples. They were then incubated at 25 ℃ for 7 days before starting the experiments, to allow any early sampling and sieving effects to subside ".

3.  The source of the unstable substrate is not clearly stated in the Conclusion, and the role of cotton meal regulation needs to be more clearly emphasised; and future research directions are not proposed;

    **Response**:We have add this sentence following your suggestion.We revise the conclusion : " Cotton meal is a kind of organic material with high nitrogen content, adding cotton meal in salinised soil can stimulate and promote the release of soil nutrients. The microorganisms mainly use the organic matter in the cotton meal in the pre-culture period, so the soil carbon excitation is negative excitation, Soil priming effect turned from negative to positive at the later stage of incubation (day 28), because microorganisms turned to decompose SOC from the labile substrate. With the increase of salinity, the diversity of microbial community decreased. Soil microbial community was mainly controled by soil pH and EC. By O2PLS, we found Actinobacteria and Proteobacteria (Luteimonas, Hoeflea and Stenotrophomonas) dominant in these soils were the core microbial taxa that affecting the process of organic C mineralization, particularly soil primed $CO_2$ ".

4.  The first occurrence of a term in line 184 manuscript should be given the full name. Also, in line 399, nouns should be given full names, and the inappropriate use of abbreviations could be improved;

    **Response**: We have added the explanations for " PDB " following your suggestion. Please see lines 184 "where Rsample is the mass ratio of $^{13}C$ to $^{12}C$ of each sample and RVPDB is the international PDB(Peedee Belemnite) limestone standard. The labeled $^{13}C$ (%) of cottonseed meal was then estimated from ".

5.  Fig.4 The figure notes are not continued on the pictures;

    **Response:** We have revised this sentence following your suggestion. The changes are as follows  "Fig. 4. The top 10 of phylum and gene in bacterial community in soils with a gradient of salinity "

6.  The article needs to be formatted in a uniform way. For example, line112 needs a space at the beginning, line192 needs to be deleted, line250 "CO2" needs to be changed to the format of a subscript, line269 (Fig. 3), line273 needs to be deleted, and so on, and a lot of details need to be paid attention to.

    **Response**: Thank you for your suggestion. We Add a space at the beginning of the paragraph in line 112, and delete lines 192 and 269, besides change the line 250 from "CO2" to "$CO_2$". In addition, the entire manuscript was checked for details and changes were made to address formatting issues.

---

## Referee Comment (RC3)

Thank you for your advice. I will answer and revise the following questions.

1.  The concluding sentence should be more specific, e.g., "shifted dominant microorganisms", with specific species and modes of species transformation.

    **Response**:We have revised this sentence following your suggestion. Based on the reviewer's comments, add the dominant microorganisms to the conclusion and change to "Here, we highlight that the increase of salinity reduced the diversity of microbial community and shifted dominant microorganisms(Actinobacteria and Proteobacteria (Luteimonas, Hoeflea and Stenotrophomonas)) that determined SOC priming effects, which provides a theoretical basis for understanding of SOC dynamics and microbial drivers under salinity gradient. "

2.  The order of the preface should be adjusted by placing the current status and hazards of soil salinization in the first paragraph, followed by a description of soil organic carbon and soil initiation effects, and their effects on soil flora, in the context of the soil salinization problem.

    **Response**: We have modified the Introduction according to your comment. We have re-adjusted the order of the Introduction.

3.  Line 45: References are incorrectly cited in format.

    **Response**: We have corrected it in the Manuscript. The format of the reference in line 45 has been changed. "The input of substrate C can influence the output (i.e., $CO_2$ release) through a phenomenon called priming effect, which was firstly discovered by LÖhnis (1926). "

4.  Is it inter-root or non-root soil? Please add it.

    **Response**:We have revised this sentence following your suggestion. The soil samples is non-rhizosphere soil.Please see lines 96-100"In September 2021, we determining the sampling area, and use the five-point sampling method to collecting non-rhizosphere soil. The soil samples were indoor air drying and hand-picked to remove visible other debris, animal and plant residues and then sieved at field moisture (<2mm) and subsequently adjusted to 40% of water holding    capacity (WHC). Texture was determined by the pipette method without carbonate in all soil samples. They were then incubated at 25 ℃ for 7 days before starting the experiments, to allow any early sampling and sieving effects to subside. "

5.  Line 122: Soil: water should be 1:2.5 for the determination of soil pH and EC. Please refine the methods for the determination of soil total carbon and nitrogen content, soil microbial carbon and soil organic carbon.

**Response:**We have revised this sentence following your suggestion. In this experiment, soil pH and EC were determined by the method of Bao (2000), and your comments will be taken into account to improve the test method in future tests. And the methods of soil total carbon and nitrogen content determination were improved. Please see lines 115-131  "Air-dry soil (5 g, <2 mm) and 25 ml of deionised water were shaken together for 1 min and left to settle for 30 min, which was repeated once more before pH was determined with a pH electrode. Soil water-soluble salt was analyzed by weighted at a soil:water ratio of 1:5 (weight/weight). Air-dry soil (5 g, <1 mm) and 25 ml of deionised water were shaken together for 30 min, filtration to obtain clear filtrate, using thermostat water bath to evaporate and weigh(Bao, 2000). Soil total carbon (TC), total nitrogen (TN) are collect soil to be tested was dried and ground through a 0.15mm screen, and a certain amount of treated soil sample was wrapped in tin foil and placed in an element analyzer for determinatio (air-dried, milled <150 μm) were determined by dry combustion (LECO CNS 2000, LECO Corporation, Michigan, USA). Soil microbial biomass C was determined by fumigation extraction (Vance et al., 1987; Wu et al., 1990) ".

6.    Line 250: Please revise the "CO2" subscript.

    **Response**: Sorry for this error.We revise the "CO2" to "$CO_2$".

7.    Line 377: Please keep the italics of the bacterial colony consistent.

    **Response**:We have revised this sentence following your suggestion. We have examined slants of full-text bacterial colonies with consistency.

8.    The conclusion section lacks content related to the stimulation of soil salinity by cotton meal and its regulation of the distribution of bacterial flora, and should focus on the role of cotton meal treatments on soil changes, thus highlighting the characteristics of this study.

    **Response**: Based on the opinions of all reviewers, we have adjusted the conclusion.We revise the conclusion : "Cotton meal is a kind of organic material with high nitrogen content, adding cotton meal in salinised soil can stimulate and promote the release of soil nutrients. The microorganisms mainly use the organic matter in the cotton meal in the pre-culture period, so the soil carbon excitation is negative excitation, Soil priming effect turned from negative to positive at the later stage of incubation (day 28), because microorganisms turned to decompose SOC from the labile substrate. With the increase of salinity, the diversity of microbial community decreased. Soil microbial community was mainly controled by soil pH and EC. By O2PLS, we found Actinobacteria and Proteobacteria (Luteimonas, Hoeflea and Stenotrophomonas) dominant in these soils were the core microbial taxa that affecting the process of organic C mineralization, particularly soil primed $CO_2$ ".

---

## Referee Comment (RC4)

[referee-annotated manuscript omitted]

---

## Referee Comment (RC5)

[referee-annotated manuscript omitted]

---

## Author Response (AR1)

**Author's response to referee 1**

Thank you for your advice. I will answer and revise the following questions.

1.  Materials and methods section 2.1 Soil sampling and cottonseed meal production does not provide a detailed description of the sampling site and does not mention the characteristics of the local soil type, etc. Please add.

    **Response:** Many thanks for your review and suggestions. Please see lines 96-100 "The soil type was gray desert soil, which was collected from farmlands (82.90° longitude, 44.96° latitude) in Xiao Yinpan town,Bole City, Bortala, northern Xinjiang Uygur Autonomous Region, northwest China. The farmlands soil is naturally formed original saline-salinity soil and with a continuous 30 years planting of maize (C4 crop) and maize straw returning to soil for 7-8 year. " And the cottonseed meal productions see the lines 106-109 and table 1.

2.  What is the season for the collection of agricultural soil samples? Is it an inter-root or non-root soil? What is the soil's mode of collection? Please add.

    **Response:** We have revised this sentence following your suggestion. Please see lines 101-107 "In September 2021, we determining the sampling area, and use the five-point sampling method to collecting non-rhizosphere soil.  The soil samples were indoor air drying and hand-picked to remove visible other debris, animal and plant residues and then sieved at field moisture (<2mm) and subsequently adjusted to 40% of water holding  capacity (WHC). Texture was determined by the pipette method without carbonate in all soil samples. They were then incubated at 25 ℃ for 7 days before starting the experiments, to allow any early sampling and sieving effects to subside ".

3.  The source of the unstable substrate is not clearly stated in the Conclusion, and the role of cotton meal regulation needs to be more clearly emphasised; and future research directions are not proposed;

    **Response:**We have add this sentence following your suggestion.We revise the conclusion : " Cotton meal is a kind of organic material with high nitrogen content, adding cotton meal in salinised soil can stimulate and promote the release of soil nutrients. The microorganisms mainly use the organic matter in the cotton meal in the pre-culture period, so the soil carbon excitation is negative excitation, Soil priming effect turned from negative to positive at the later stage of incubation (day 28), because microorganisms turned to decompose SOC from the labile substrate. With the increase of salinity, the diversity of microbial community decreased. Soil microbial community was mainly controled by soil pH and EC. By O2PLS, we found Actinobacteria and Proteobacteria (Luteimonas, Hoeflea and Stenotrophomonas) dominant in these soils were the core microbial taxa that affecting the process of organic C mineralization, particularly soil primed $CO_2$ ".

4. The first occurrence of a term in line 184 manuscript should be given the full name. Also, in line 399, nouns should be given full names, and the inappropriate use of abbreviations could be improved;

   **Response**: We have added the explanations for " PDB " following your suggestion. Please see lines 184 "where Rsample is the mass ratio of $^{13}C$ to $^{12}C$ of each sample and RVPDB is the international PDB(Peedee Belemnite) limestone standard. The labeled $^{13}C$ (%) of cottonseed meal was then estimated from ".

5. Fig.4 The figure notes are not continued on the pictures;

   **Response:** We have revised this sentence following your suggestion. The changes are as follows  "Fig. 4. The top 10 of phylum and gene in bacterial community in soils with a gradient of salinity "

6. The article needs to be formatted in a uniform way. For example, line112 needs a space at the beginning, line192 needs to be deleted, line250 "CO2" needs to be changed to the format of a subscript, line269 (Fig. 3), line273 needs to be deleted, and so on, and a lot of details need to be paid attention to.

   **Response**: Thank you for your suggestion. We Add a space at the beginning of the paragraph in line 112, and delete lines 192 and 269, besides change the line 250 from "CO2" to "$CO_2$". In addition, the entire manuscript was checked for details and changes were made to address formatting issues.

**Author's response to referee2**

Thank you for your advice. I will answer and revise the following questions.

1.  The concluding sentence should be more specific, e.g., "shifted dominant microorganisms", with specific species and modes of species transformation.

    **Response**:We have revised this sentence following your suggestion. Based on the reviewer's comments, add the dominant microorganisms to the conclusion and change to "Here, we highlight that the increase of salinity reduced the diversity of microbial community and shifted dominant microorganisms(Actinobacteria and Proteobacteria (Luteimonas, Hoeflea and Stenotrophomonas)) that determined SOC priming effects, which provides a theoretical basis for understanding of SOC dynamics and microbial drivers under salinity gradient. "

2.  The order of the preface should be adjusted by placing the current status and hazards of soil salinization in the first paragraph, followed by a description of soil organic carbon and soil initiation effects, and their effects on soil flora, in the context of the soil salinization problem.

    **Response**: We have modified the Introduction according to your comment. We have re-adjusted the order of the Introduction.

3.  Line 45: References are incorrectly cited in format.

    **Response**: We have corrected it in the Manuscript. The format of the reference in line 45 has been changed. "The input of substrate C can influence the output (i.e., $CO_2$ release) through a phenomenon called priming effect, which was firstly discovered by LÖhnis (1926). "

4.  Is it inter-root or non-root soil? Please add it.

    **Response**:We have revised this sentence following your suggestion. The soil samples is non-rhizosphere soil.Please see lines 96-100"In September 2021, we determining the sampling area, and use the five-point sampling method to collecting non-rhizosphere soil. The soil samples were indoor air drying and hand-picked to remove visible other debris, animal and plant residues and then sieved at field moisture (<2mm) and subsequently adjusted to 40% of water holding capacity (WHC). Texture was determined by the pipette method without carbonate in all soil samples. They were then incubated at 25 ℃ for 7 days before starting the experiments, to allow any early sampling and sieving effects to subside. "

5. Line 122: Soil: water should be 1:2.5 for the determination of soil pH and EC. Please refine the methods for the determination of soil total carbon and nitrogen content, soil microbial carbon and soil organic carbon.

**Response:**We have revised this sentence following your suggestion. In this experiment, soil pH and EC were determined by the method of Bao (2000), and your comments will be taken into account to improve the test method in future tests. And the methods of soil total carbon and nitrogen content determination were improved. Please see lines 115-131 "Air-dry soil (5 g, <2 mm) and 25 ml of deionised water were shaken together for 1 min and left to settle for 30 min, which was repeated once more before pH was determined with a pH electrode. Soil water-soluble salt was analyzed by weighted at a soil:water ratio of 1:5 (weight/weight). Air-dry soil (5 g, <1 mm) and 25 ml of deionised water were shaken together for 30 min, filtration to obtain clear filtrate, using thermostat water bath to evaporate and weigh(Bao, 2000). Soil total carbon (TC), total nitrogen (TN) are collect soil to be tested was dried and ground through a 0.15mm screen, and a certain amount of treated soil sample was wrapped in tin foil and placed in an element analyzer for determinatio (air-dried, milled <150 μm) were determined by dry combustion (LECO CNS 2000, LECO Corporation, Michigan, USA). Soil microbial biomass C was determined by fumigation extraction (Vance et al., 1987; Wu et al., 1990) ".

6. Line 250: Please revise the "CO2" subscript.

**Response**: Sorry for this error.We revise the "CO2" to "$CO_2$".

7. Line 377: Please keep the italics of the bacterial colony consistent.

**Response**:We have revised this sentence following your suggestion. We have examined slants of full-text bacterial colonies with consistency.

8. The conclusion section lacks content related to the stimulation of soil salinity by cotton meal and its regulation of the distribution of bacterial flora, and should focus on the role of cotton meal treatments on soil changes, thus highlighting the characteristics of this study.

**Response**: Based on the opinions of all reviewers, we have adjusted the conclusion.We revise the conclusion : "Cotton meal is a kind of organic material with high nitrogen content, adding cotton meal in salinised soil can stimulate and promote the release of soil nutrients. The microorganisms mainly use the organic matter in the cotton meal in the pre-culture period, so the soil carbon excitation is negative excitation, Soil priming effect turned from negative to positive at the later stage of incubation (day 28), because microorganisms turned to decompose SOC from the labile substrate. With the increase of salinity, the diversity of microbial community decreased. Soil microbial community was mainly controled by soil pH and EC. By O2PLS, we found Actinobacteria and Proteobacteria (Luteimonas, Hoeflea and Stenotrophomonas) dominant in these soils were the core microbial taxa that affecting the process of organic C mineralization, particularly soil primed $CO_2$ ".

9. Please confirm the correct format for the use of i.e. and standardize the usage in the manuscript.

   **Response:** Thank you for your suggestion. After checking the "i.e. " throughout the manuscript, we have changed the "i.e." to "i.e.," in lines 23 and 55 of the manuscript.

10. Please modify the reference citation format in lines 61, 64, and 67-68 of the manuscript to correct.

    **Response:** Many thanks for your review and suggestions. We have modify the reference citation format of the full manuscript references.

11. Geographical descriptions are made without indicating east-west longitude, north-south. Please add a specific description on Line 97.

    **Response:** We have revised this sentence following your suggestion.We have changed " The soil type was gray desert soil, which was collected from farmlands (82.90° longitude, 44.96° latitude) in Xiao Yinpan town, Bole City, Bortala, northern Xinjiang Uygur Autonomous Region, northwest China. " as " The soil type was gray desert soil, which was collected from farmlands (82.90° E, 44.96° N) in Xiao Yinpan town, Bole City, Bortala, northern Xinjiang Uygur Autonomous Region, northwest China. "

12. Please clarify in section 2.3 the salt gradient that explains how it was determined.

    **Response:**We have add this sentence following your suggestion. We add the sentence "Determination of five salinity gradients at 0.25%, 0.58%, 0.75%, 1.00% and 2.64% through soil salinity measurements. " in the line104 -106. And we also add the sentence " salinity gradient (salinity at 0.25%, 0.58%, 0.75%, 1.00% and 2.64%)" in the line143-144.

13. Please make sure that all genus names and species additions in the manuscript are in italics.

    **Response:** Many thanks for your review and suggestions. We have changed all genus names and newly added species names in the manuscript to italics

14. The format of the references is not uniform, so please check and confirm one by one.

    **Response:** Thank you for your suggestion. We have check and confirm all the references in the manuscript.

15. The format of the manuscript is not uniform, please check and confirm one by one.

    **Response:** Thank you for your suggestion. We have check and confirm all the format in the manuscript.